# Prototypical Cross-Attention Networks for Multiple Object Tracking and Segmentation

**Lei Ke**[1,2]   **Xia Li**[1]   **Martin Danelljan**[1]   **Yu-Wing Tai**[3]   **Chi-Keung Tang**[2]   **Fisher Yu**[1]

[1]ETH Zürich         [2]HKUST         [3]Kuaishou Technology

{lkeab,cktang}@cse.ust.hk, {xia.li,martin.danelljan}@vision.ee.ethz.ch

yuwing@gmail.com, i@yf.io

## Abstract

Multiple object tracking and segmentation requires detecting, tracking, and segmenting objects belonging to a set of given classes. Most approaches only exploit the temporal dimension to address the association problem, while relying on single frame predictions for the segmentation mask itself. We propose **P**rototypical **C**ross-**A**ttention **N**etwork (**PCAN**), capable of leveraging rich spatio-temporal information for online multiple object tracking and segmentation. PCAN first distills a space-time memory into a set of prototypes and then employs cross-attention to retrieve rich information from the past frames. To segment each object, PCAN adopts a prototypical appearance module to learn a set of contrastive foreground and background prototypes, which are then propagated over time. Extensive experiments demonstrate that PCAN outperforms current video instance tracking and segmentation competition winners on both Youtube-VIS and BDD100K datasets, and shows efficacy to both one-stage and two-stage segmentation frameworks. Code and video resources are available at http://vis.xyz/pub/pcan.

## 1   Introduction

Multiple object tracking and segmentation (MOTS), also known as Video Instance Segmentation (VIS), is an important problem with many real-world applications, including autonomous driving [10, 26] and video analysis [4, 46]. The task involves tracking and segmenting all objects within a video from a given set of semantic classes. We are witnessing rapidly growing research interest on MOTS thanks to the introduction of large scale benchmarks [46, 50, 37]. State-of-the-art methods [46, 5, 37, 29] for MOTS mainly follow the tracking-by-detection paradigm, where objects are first detected and segmented in individual frames and then associated over time.

Although methods based on the popular tracking-by-detection philosophy have shown promising results, temporal modeling is limited to the object association phase [46, 5, 22] and only between two adjacent frames [37, 18]. On the other hand, the temporal dimension carries rich information about the scene. The information encoded in multiple temporal views of an object has the potential of improving the quality of predicted segmentation, localization, and categories. However, effectively and efficiently leveraging the rich temporal information remains a challenge. While sequential modeling has been applied for video processing [40, 41, 9, 28, 12], these methods generally operate directly on the high-resolution deep features, requiring large computational and memory consumption, which greatly limits their use.

We propose a **P**rototypical **C**ross-**A**ttention **M**odule, termed **PCAM**, to leverage temporal information for multiple object tracking and segmentation. As illustrated in Figure 1, the module first distills spatio-temporal information into condensed prototypes using clustering based on Expectation Maximization. The resulting prototypes, composed of Gaussian Components, yield a rich and generalizable yet compact representation of the past visual features. Given a deep feature embedding of the current

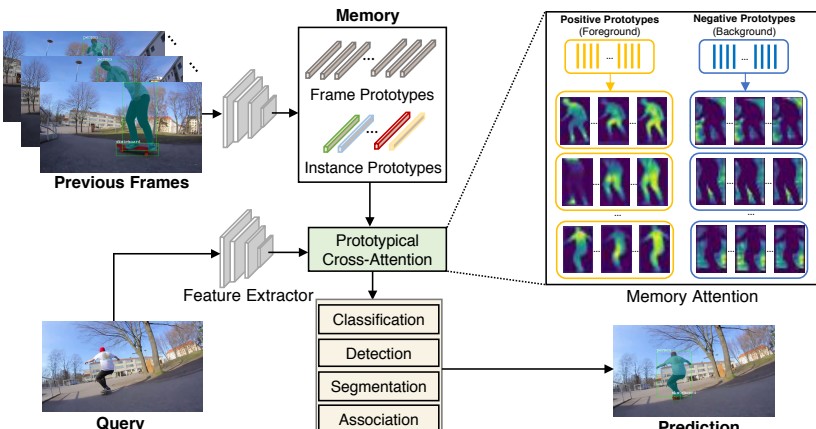

Figure 1: We propose Prototypical Cross-Attention Network for MOTS, which first condenses the space-time memory and high-resolution frame embeddings into frame-level and instance-level prototypes. These are then employed to retrieve rich temporal information from past frames by our efficient prototypical cross-attention operation.

frame, PCAM then employs prototypical cross-attention to read relevant information from prior frames.

Based on the noise-reduced clustered video features information, we further develop a **P**rototypical **C**ross-**A**ttention **N**etwork (**PCAN**) for MOTS, that integrates the general PCAM at two stages in the network: on the frame-level and instance-level. The former reconstructs and aligns temporal past frame features with current frame, while the instance level integrates specific information about each object in the video. For robustness to object appearance change, PCAN represents each object instance by learning sets of contrastive foreground and background prototypes, which are propagated in an online manner. With a limited number of prototypes for each instance or frame, PCAN efficiently performs long-range feature aggregation and propagation in a video with linear complexity. Consequently, our PCAN outperforms standard non-local attention [40] and video transformer [41] on both the large-scale Youtube-VIS and BDD100K MOTS benchmarks.

Our main contributions are summarized as follows: (i) We introduce the PCAN module for efficiently utilizing long-term spatio-temporal video information. (ii) We develop a MOTS approach that employs PCAN on frame and instance-level. (iii) We further represent the appearance of each video tracklet with contrastive foreground and background prototypes, which are propagated over time. (iv) We extensively analyze our approach. Our PCAN outperforms previous approaches on the challenging self-driving dataset BDD100K [50] and the semantically diverse YouTube-VIS dataset [46].

## 2 Related work

**Video instance segmentation (VIS)** Existing VIS methods [46, 2, 21] widely adapt the two-stage paradigm of Mask R-CNN [11] and its variants [13, 15] by adding an additional tracking branch. Thus, their typical pipelines first detect regions of interest (RoIs) and then use the instance features after RoIAlign to regress object mask and associate cross-frame instances. More recent works [5, 18, 22, 48] employ a one-stage instance segmentation method, e.g. the anchor-free FCOS detector [34], which predicts a linear combination of mask bases [3] as its final segmentation. The aforementioned approaches make very limited use of temporal information to enhance the quality of the segmentation, instead relying on single image-based mask prediction, or only model short-term temporal correlation between two consecutive frames [18, 30]. In the context of long-term temporal association, the offline method VisTr [41] adapts vision transformer [6] for VIS, but suffers from a huge computational burden and memory consumption due to the dense pixel-level attention operations over long sequences. Compared to these methods, our PCAN temporally aggregates and propagates the prototypical features with both the long-term benefit and linear complexity.

**Multiple Object Tracking and Segmentation (MOTS)** Similar to VIS, MOTS methods [37, 27, 29] mainly follow the tracking-by-detection paradigm. Objects are first detected and segmented, followed

by association between frames. Track R-CNN [37] integrates temporal context feature from two neighboring frames using 3D convolutions. TrackFormer [25] performs joint object detection and tracking by recurrently using Transformers, while Stem-Seg [1] adopts a short 3D convolutional spatio-temporal volume to learn pixel embedding by treating segmentation as a bottom-up grouping. In contrast, our approach clusters appearance features in a long spatio-temporal volume with explicit foreground and background prototypes that are updates online. Besides, the mixture Gaussian components in instance appearance module equips PCAN a stronger modeling ability compared to instance-level average pooling [33, 49] or single Gaussian model [51, 14].

**Temporal attention models** Video understanding usually requires long-range sequential modeling of relations between spatio-temporal locations. Recently, attention-based approaches, such as non-local attention [40, 39, 28, 12] and transformers [8, 35, 16], have been successfully adopted in video classification and action recognition. These tasks [23, 32, 43] involve dense pixel-level attention, leading to quadratic complexity in the sequence length, thus making them excessively expensive for long sequences. Improved temporal attention models mainly include double attention mechanism [7] on image recognition with global-local decomposition, and clustered attention Transformer [38] for language sequence modeling. Besides, recent prototypical methods [19, 45] use the EM algorithm for single-image semantic segmentation or few-shot learning [33]. Unlike these methods, our PCAN uses compact prototypical representation both for temporal feature aggregation and compact instance appearance feature propagation.

# 3 Method

We propose an approach for Multiple Object Tracking and Segmentation. Given a video sequence, the goal is to detect, track, and segment objects from a predefined set of object categories. Specifically, we consider the online setting, where the predictions only depend on current and past frames.

## 3.1 Traditional Cross-Attention

To utilize the rich temporal information to improve the segmentation prediction, recent approaches [28, 12] have employed cross-attention. We consider past spatio-temporal information encoded in a memory $\mathbf{M}$, consisting of deep features of size $H \times W \times T \times C$. The memory encapsulates valuable information about the past appearances and predictions of objects and background in a scene. To attend to the memory, the information is first separately embedded into key $\mathbf{k}^M$ and value $\mathbf{v}^M$ feature vectors. The keys are used to address relevant memories whose corresponding values are returned. The standard memory reading process is a non-local operation computed as the weighted sum,

$$y_i = \frac{1}{Z_i} \sum_{j=1}^{H \times W \times T} \exp(\mathbf{k}_i^Q \cdot \mathbf{k}_j^M) \mathbf{v}_j^M \,, \tag{1}$$

where $\mathbf{k}^Q$ denotes query key map, which is predicted from the current frame. Further, $i$ and $j$ are the index of each query and the memory location, and $Z_i = \sum_j \exp(\mathbf{k}_i^Q \cdot \mathbf{k}_j^M)$ is the normalizing factor.

Although proven effective, the standard attention operation (1) is known to suffer from poor computational and memory scaling properties [20]. In particular, since all queries are matched to all keys, it experiences a quadratic scaling $\mathcal{O}((HW)^2)$ of computations in the spatial size $HW$ of the feature map. This is particularly problematic for segmentation tasks, where fine-grained high-resolution information is desired to improve the quality of the predictions.

## 3.2 Prototypical Cross-Attention

To address the aforementioned limitations of the standard cross-attention, we introduce the prototypical cross-attention to first condense sets of high-resolution feature vectors in the past frames. Our approach is based on a clustered memory $\mathbf{M}_c$. We call these clusters prototypes, since they correspond to representative items in the memory. While clustering effectively reduces the number of items in the memory, it also serves to deprecate noisy information, leading to a more generalizable and robust representation of the memory.

To employ an attention mechanism, similar to (1), we require a clustering of the memory that generates a principled continuous and differentiable clustering assignment function. We therefore

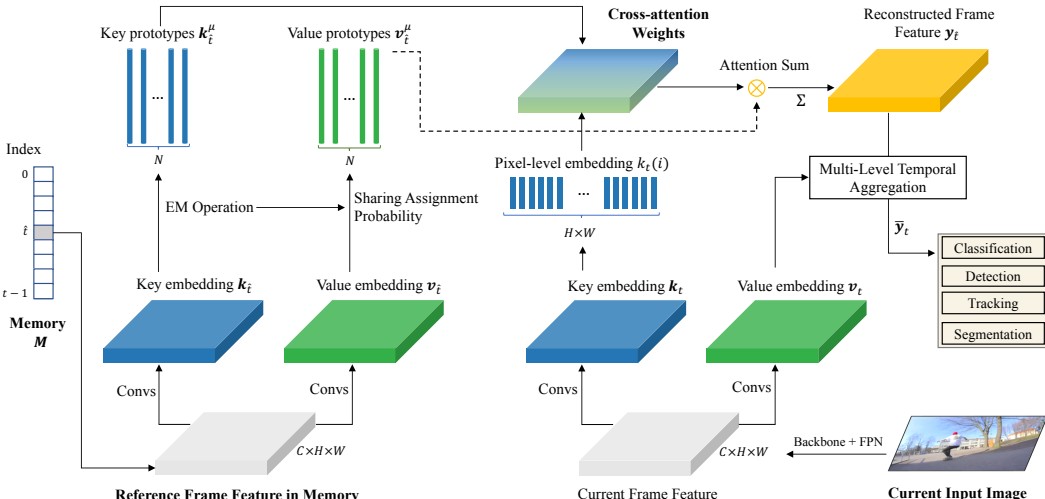

Figure 2: Overview of our frame-level prototypical cross-attention. For a frame $\hat{t}$ in the memory we first perform GMM-based clustering to achieve the key $\mathbf{k}_{\hat{t}j}^{\mu}$ and value $\mathbf{v}_{\hat{t}j}^{\mu}$ prototypes. Given the key encoding $\mathbf{k}_t$ of the current frame, we attend to the prototypes to generate the reconstructed feature $\mathbf{y}_{\hat{t}}$, which are then aggregated temporally and fused with the current value encoding $\mathbf{v}_t$.

cluster the keys in the memory by fitting a Gaussian Mixture Model (GMM),

$$p(\mathbf{k}) = \frac{1}{N} \sum_{j=1}^{N} p(\mathbf{k}|z=j), \qquad p(\mathbf{k}|z=j) = \frac{1}{(2\pi\sigma^2)^{\frac{D}{2}}} \exp\left(-\frac{1}{2\sigma^2}\|\mathbf{k} - \mathbf{k}_j^{\mu}\|^2\right) \qquad (2)$$

Here, $N$ denotes the number of Gaussian mixtures, $D$ is the feature dimension of the keys. We use a constant variance parameter $\sigma^2$ and uniform cluster priors $p(z=j) = \frac{1}{N}$, where $z$ denotes the latent cluster assignment variable. The component means $\mathbf{k}^{\mu}$ represent the prototype keys in the memory. We generate the clustering (2) using the standard Expectation-Maximization algorithm.

The GMM allows us to compute a soft cluster assignment by evaluating the posterior probability of the latent assignment variable $z$. Using Bayes rule, the probability of a key value $\mathbf{k}$ to be assigned to the $j$th prototype is derived as,

$$p(z=j|\mathbf{k}) = \frac{p(\mathbf{k}|z=j)p(z=j)}{\sum_{l=1}^{N} p(\mathbf{k}|z=l)p(z=l)} = \frac{\exp\left(-\frac{1}{2\sigma^2}\|\mathbf{k} - \mathbf{k}_j^{\mu}\|^2\right)}{\sum_{l=1}^{N} \exp\left(-\frac{1}{2\sigma^2}\|\mathbf{k} - \mathbf{k}_l^{\mu}\|^2\right)}. \qquad (3)$$

The resulting cluster assignment can thus be written as a SoftMax operation, where the corresponding logits are provided by the negative cluster distance $\|\mathbf{k} - \mathbf{k}_j^{\mu}\|^2$ scaled with a temperature of $2\sigma^2$.

Since the clustering is performed in the key space of the memory, we next retrieve the corresponding value prototypes. To this end, we employ the key cluster assignment probabilities in (3) to compute the values for each memory prototype,

$$\mathbf{v}_j^{\mu} = \sum_{l=1}^{H \times W} p(z=j|\mathbf{k}_l^{M})\mathbf{v}_l^{M}. \qquad (4)$$

For attending to our clustered memory, we first predict the key encodings $\mathbf{k}_i^{Q}$ of the query image. We then read from the clustered memory by computing the average over the value prototypes $\mathbf{v}_j^{\mu}$, weighted with the cluster assignment probabilities,

$$\mathbf{y}_i = \sum_{j=1}^{N} p(z=j|\mathbf{k}_i^{Q})\mathbf{v}_j^{\mu} = \frac{1}{Z_i} \sum_{j=1}^{N} \exp\left(-\frac{1}{2\sigma^2}\|\mathbf{k}_i^{Q} - \mathbf{k}_j^{\mu}\|^2\right) \mathbf{v}_j^{\mu}. \qquad (5)$$

The final attention operation has much similarity with the original dot-product cross attention (1). Note that the key-query similarity in our approach is measured by Euclidian distance instead of a dot-product. Importantly, our formulation (5) attends to a reduced set of $N$ prototypes, while the original attention (1) requires attending to the full spatio-temporal memory of size $H \times W \times T$.

### 3.3 Prototypical Cross-Attention Network

Here, we propose the Prototypical Cross-Attention Network (PCAN) for MOTS by integrating our prototypical cross-attention module into both the frame-level and instance-level. The former aims to align and aggregate temporal frame features stored in memory, while the latter is for propagating the instance appearance features over time and produce instance cross-attention maps to help segmentation. Besides, we also design a prototypical instance appearance module to represent each video tracklet with contrastive mixture foreground and background prototypes.

#### 3.3.1 Frame-level Prototypical Cross-Attention

In Figure 2, prototypical cross-attention first produces prototypes by fitting a Gaussian mixtures model (2) to the feature in the memory. To provide further flexibility when dynamically updating the memory $\mathbf{M}$, we first perform frame-wise clustering for each reference frame feature at index $\hat{t}$ to compute the $N$ key prototypes $\{\mathbf{k}_{\hat{t}i}^{\mu}\}_{j=1}^{N}$, and retrieve the corresponding value embeddings $\{\mathbf{v}_{\hat{t}j}^{\mu}\}_{j=1}^{N}$ using (4) for each memory frame $\hat{t}$ independently. The key and value features are predicted using two parallel convolutional layers.

**Frame-wise prototypical memory attention** Given the query key encoding $\mathbf{k}_{ti}^{Q}$ of the current frame $t$, we perform prototypical cross-attention to each memory frame $\hat{t}$ independently using our formulation (3) as,

$$\mathbf{y}_{\hat{t}i} = \frac{1}{Z_{\hat{t}i}} \sum_{j=1}^{N} \exp\left(-\frac{1}{2\sigma^2}\|\mathbf{k}_{ti} - \mathbf{k}_{\hat{t}j}^{\mu}\|^2\right) \mathbf{v}_{\hat{t}j}^{\mu}, \qquad Z_{\hat{t}i} = \sum_{l=1}^{N} \exp\left(-\frac{1}{2\sigma^2}\|\mathbf{k}_{ti}^{Q} - \mathbf{k}_{\hat{t}l}^{\mu}\|^2\right). \quad (6)$$

Note that the index $i$ refers to a spatial coordinate in the current frame. The resulting feature map $\mathbf{y}_{\hat{t}}$ can intuitively be seen as a projection of features from frame $\hat{t}$ to the current frame. This projection essentially aligns the condensed feature information in frame $\hat{t}$ with the current frame.

**Temporal feature aggregation** Since frame-wise attention does not fuse temporal information, we perform a temporal aggregation. The temporal information $\mathbf{y}_{\hat{t}}$ in (6) from different frames $\hat{t}$ are fused as a linear combination, weighted by the feature similarity with the current frame. Specifically, the temporally aggregated representation is obtained as

$$\bar{\mathbf{y}}_{ti} = \sum_{\hat{t}=1}^{t} w_{\hat{t}i}\mathbf{y}_{\hat{t}i}, \qquad w_{\hat{t}i} = \frac{\exp(\mathbf{y}_{ti} \cdot \mathbf{y}_{\hat{t}i})}{\sum_{s=1}^{t} \exp(\mathbf{y}_{ti} \cdot \mathbf{y}_{si})}. \qquad (7)$$

Note that $\hat{t} = t$ in the sum refers to the value embedding $\mathbf{y}_{ti} = \mathbf{v}_{ti}^{Q}$ extracted from the current frame. The contribution of each frame $\hat{t}$ is thus weighted by the similarity to this current frame prediction using the attention weights $w_{\hat{t}i}$. This strategy ensures that incorrect or dissimilar regions are suppressed when computing the final aggregated feature embedding $\bar{\mathbf{y}}_t$. To handle object with large-scale variation and produce more fine-grained instance mask prediction, we further extend temporal aggregation to multi-level using different levels of the extracted FPN features, as detailed in the supplementary material.

#### 3.3.2 Instance-level Prototypical Cross-Attention

**Contrastive foreground and background representation** In additional to the condensed frame-level representation, for more accurate segmentation results, we further encode each tracked object with compact and robust appearance prototypes. To further empower our proposed attention mechanism, we utilize the initially detected object mask to identify each foreground instance. We then separately model the extracted foreground and background features using a GMM (2). We denote the resulting foreground prototypes as $\mathbf{k}_{tj}^{+}$ and background prototypes as $\mathbf{k}_{tj}^{-}$. The former thus focuses on the appearance of the specific object, creating a rich and dynamic appearance model. When employed in our prototypical cross-attention framework (Section 3.2), it provides fine-grained attention from localized prototypes that naturally learn to focus specific parts of views of the object, as visualized in Fig. 3. Furthermore, the background prototypes $\mathbf{k}_{tj}^{-}$ capture valuable information about the background appearance, which can greatly alleviate the segmentation process. For each object instance we attend to the foreground and background prototypes separately using (3). The results are

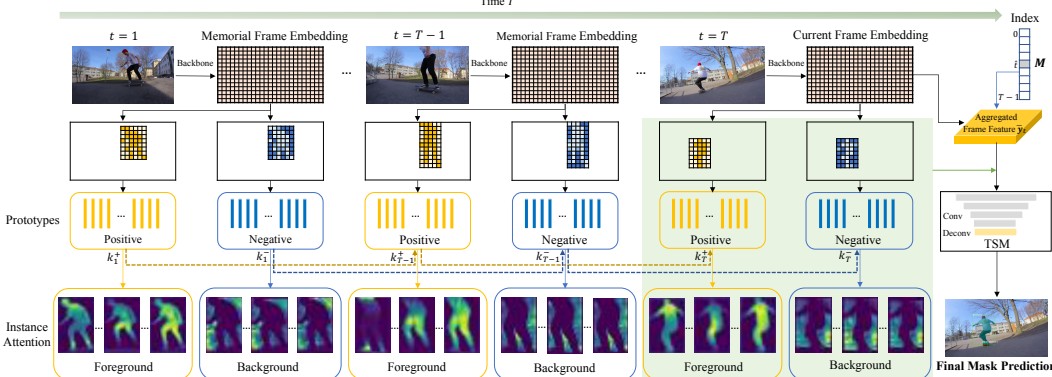

Figure 3: Our instance-level prototypical attention with foreground and background prototypes and temporal propagation. The foreground/background attention maps from (bottom) demonstrate the localized and discriminative appearance representation. Temporal Segmentation Module (TSM) takes the current frame, initial mask, and instance attention maps as input and generates the final mask.

concatenated together with the initial mask detection to the Temporal Segmentation Head (TSM) for final prediction, as illustrated in Figure 3.

**Tracklet feature propagation and updating** To effectively model the object appearance change and preserve the most relevant information, we design a recurrent instance appearance updating scheme. From the first video frame where object appears, the accumulated prototypes $\bar{\mathbf{k}}_{tj}^{+}$, $\bar{\mathbf{k}}_{tj}^{-}$ for the instance are propagated to the subsequent frames and updated with new appearance prototypes $\mathbf{k}_{tj}^{+}$, $\mathbf{k}_{tj}^{-}$ using an update rate $\lambda$ as,

$$\bar{\mathbf{k}}_{tj}^{+} = (1-\lambda)\bar{\mathbf{k}}_{t-1,j}^{+} + \lambda\mathbf{k}_{tj}^{+}, \qquad \bar{\mathbf{k}}_{tj}^{-} = (1-\lambda)\bar{\mathbf{k}}_{t-1,j}^{-} + \lambda\mathbf{k}_{tj}^{-}. \tag{8}$$

Figure 3 also reveals the consistency of the attended region of a specific prototype $j$.

## 4 Experiments

Here, we present comprehensive evaluation and analysis of our approach. Experiments are performed on two large scale datasets, namely YouTube-VIS [46] and BDD100K [50].

### 4.1 Experiment setup

**Youtube-VIS** YouTube-VIS-2019 [46] dataset contains 2,883 high quality videos with 131k annotated object instances belonging to 40 diverse categories. The task is to simultaneously classifying, segment and track object instances belonging to these categories. The evaluation metrics for this task are an adaptation of the Average Precision (AP) and Average Recall (AR) of image instance segmentation.

**BDD100K** We also evaluate on the large-scale tracking and segmentation dataset of BDD100K [50], which is a challenging self-driving dataset with 154 videos (30,817 images) for training, 32 videos (6,475 images) for validation, and 37 videos (7,484 images) for testing. The dataset provides 8 annotated categories for evaluation, where the images in the tracking set are annotated per 5 FPS with 30 FPS frame rate. We adopt the well-established MOTS metrics [37] to our task.

**Implementation details** We implement PCAN based on two different existing MOTS approaches. For Youtube-VIS, we adopt ResNet with FPN pre-trained on COCO as the backbone, and build our segmentation tracker on the one-stage segmentation model [5]. Both the instance and frame cross-attention is built on the extracted FPN features. Our model is trained with initial learning rate 0.0025 on 4 GPUs using SGD, and executes with a speed of 15.0 FPS on ResNet-50. Similar to [46, 22, 18], we use the input size 360×640 for training. On BDD100K, we build PCAN by extending the two-stage MOT method [29] with our temporal segmentation modules. We follow the same training strategy of QDTrack-mots [29]. More details can be found in supplemental material.

Table 1: Comparison with state-of-the-art on the YouTube-VIS validation set. Results are reported in terms of mask accuracy (AP) and recall (AR). Asterisks * denote concurrent works on arXiv.

| Method | Backbone | Type | Online | AP | $AP_{50}$ | $AP_{75}$ | $AR_1$ | $AR_{10}$ |
|---|---|---|---|---|---|---|---|---|
| VisTr* [41] | ResNet-50 | Transformer | × | 35.6 | 56.8 | 37.0 | 35.2 | 40.2 |
| OSMN [47] | ResNet-50 | Two-stage | ✓ | 23.4 | 36.5 | 25.7 | 28.9 | 31.1 |
| FEELVOS [36] | ResNet-50 | Two-stage | ✓ | 26.9 | 42.0 | 29.7 | 29.9 | 33.4 |
| DeepSORT [42] | ResNet-50 | Two-stage | ✓ | 26.1 | 42.9 | 26.1 | 27.8 | 31.3 |
| MaskTrack R-CNN [46] | ResNet-50 | Two-stage | ✓ | 30.3 | 51.1 | 32.6 | 31.0 | 35.5 |
| STEm-Seg [1] | ResNet-50 | One-stage | × | 30.6 | 50.7 | 33.5 | 31.6 | 37.1 |
| SipMask [5] | ResNet-50 | One-stage | ✓ | 32.5 | 53.0 | 33.3 | 33.5 | 38.9 |
| STMask* [18] | ResNet-50 | One-stage | ✓ | 33.5 | 52.1 | 36.9 | 31.1 | 39.2 |
| SG-Net* [22] | ResNet-50 | One-stage | ✓ | 34.8 | **56.1** | 36.8 | 35.8 | 40.8 |
| **PCAN (Ours)** | ResNet-50 | One-stage | ✓ | **36.1** | 54.9 | **39.4** | **36.3** | **41.6** |
| STMask* [18] | ResNet-101 | One-stage | ✓ | 36.3 | 55.2 | 39.9 | 33.7 | 42.0 |
| SG-Net* [22] | ResNet-101 | One-stage | ✓ | 36.3 | 57.1 | 39.6 | 35.9 | 43.0 |
| **PCAN (Ours)** | ResNet-101 | One-stage | ✓ | **37.6** | **57.2** | **41.3** | **37.2** | **43.9** |

Table 2: State-of-the-art comparison on the BDD100K segmentation tracking validation set. I: ImageNet. C: COCO. S: Cityscapes. B: BDD100K. "-fix" means adopting the pretrained model from the BDD100K tracking set, fixing the existing parts, and only training the added mask head.

| Method | Pretrained | Online | mMOTSA↑ | mMOTSP↑ | mIDF↑ | ID sw.↓ | mAP↑ |
|---|---|---|---|---|---|---|---|
| SortIoU | I, C, S | ✓ | 10.3 | 59.9 | 21.8 | 15951 | 22.2 |
| MaskTrackRCNN [36] | I, C, S | ✓ | 12.3 | 59.9 | 26.2 | 9116 | 22.0 |
| STEm-Seg [1] | I, C, S | × | 12.2 | 58.2 | 25.4 | 8732 | 21.8 |
| QDTrack-mots [29] | I, C, S | ✓ | 22.5 | 59.6 | 40.8 | 1340 | 22.4 |
| QDTrack-mots-fix [29] | I, B | ✓ | 23.5 | 66.3 | 44.5 | 973 | 25.5 |
| **PCAN (Ours)** | I, B | ✓ | **27.4** | **66.7** | **45.1** | **876** | **26.6** |

## 4.2 State-of-the-Art Comparison

We compare our approach with the state-of-the-art methods on the aforementioned large-scale MOTS/VIS benchmarks Youtube-VIS and BDD100K, where PCAN outperforms all existing methods without bells and whistles, and shows efficacy to both one-stage and two-stage segmentation frameworks. We follow the official metrics of each benchmark to evaluate our model.

**Youtube-VIS** The results of Youtube-VIS benchmark is in Table 1, where PCAN achieves the best mask AP of 36.1% using ResNet-50 and 37.6% using ResNet-101 respectively, while being an online method. Our approach consistently surpasses most recent SOTA methods, including STMask [18] and SG-Net [22] by a significant margin. These methods only conduct temporal modeling between two adjacent frames for feature correlation. Compared to our baseline SipMask [5], a single-image based segmentation with object centerness association, PCAN improves the mask AP from 32.5% to 36.1%, which shows the effectiveness of long-term temporal modeling in helping object tracking and segmentation.

**BDD100K** Table 2 shows our results on BDD100K tracking and segmentation benchmark, where PCAN outperforms the strong baseline methods MaskTrackRCNN [46] and QDTrack-mots [29]. Our approach achieves a large advantage in mMOTSA, with over 3 points gain and around 10% ID switches decrease. MOTSA measures segmentation as well as tracking quality, while ID Switches can measure the performance of identity consistency. The significant advancements demonstrate that our method with prototypical cross-attention enables more accurate pixel-wise object tracking by effectively exploiting temporal information.

## 4.3 Ablation study and analysis

We conduct detailed ablation studies on Youtube-VIS validation set, where we investigate the effect of our proposed prototypical cross-attention components for MOTS during training and testing.

**Effect of frame-level prototypical cross-attention module** To study the importance of temporal information amount, we conduct an ablation study on models with different input temporal window

Table 3: Results of varying temporal memory length in our PCAN on YouTube-VIS.

| Length | AP | $AP_{50}$ | $AP_{75}$ | $AR_1$ | $AR_{10}$ |
|---|---|---|---|---|---|
| 1 | 32.5 | 53.0 | 33.3 | 33.5 | 38.9 |
| 2 | 33.7 | 53.8 | 35.3 | 33.9 | 39.5 |
| 4 | 33.9 | **54.0** | 36.8 | 34.1 | 40.0 |
| 8 | 34.2 | 53.7 | 37.6 | 34.4 | 40.3 |
| 16 | 34.6 | 53.7 | 38.3 | 35.4 | 40.5 |
| 32 | **35.4** | 53.8 | **39.1** | **35.9** | **41.0** |

Table 4: Effect of multi-layer prototypical feature fusion with tube length 4 on YouTube-VIS.

| FPN Layer | AP | $AP_{50}$ | $AP_{75}$ | $AR_1$ | $AR_{10}$ |
|---|---|---|---|---|---|
| P3 | 30.8 | 51.7 | 32.0 | 32.6 | 37.0 |
| P4 | 32.0 | 51.5 | 34.1 | 32.6 | 37.2 |
| P5 | 32.9 | 52.1 | 35.9 | 33.2 | 38.6 |
| P3-P4 | 33.1 | 52.3 | 35.6 | 33.6 | 38.5 |
| P3-P5 | **33.9** | **54.0** | **36.8** | **34.1** | **40.0** |

Table 5: Comparison with non-local attention [39] and transformer [6, 41] on YouTube-VIS.

| Length | Prototypical Cross-Attention | | | Non-local Attention | | | Transformer (Multi-Head Self-Attention) | | |
|---|---|---|---|---|---|---|---|---|---|
| | AP | FLOPs(B) | Memory(M) | AP | FLOPs(B) | Memory(M) | AP | FLOPs(B) | Memory(M) |
| 2 | 33.7 | 5.8 | 323 | 33.2 | 24.3 | 2497 | 24.6 | 103.8 | 5321 |
| 4 | 33.9 | 12.0 | 652 | 33.3 | 49.1 | 4763 | 25.8 | 387.2 | 9844 |
| 8 | **34.2** | 23.7 | 1419 | 33.6 | 99.6 | 9631 | 28.3 | 1413.3 | 18762 |

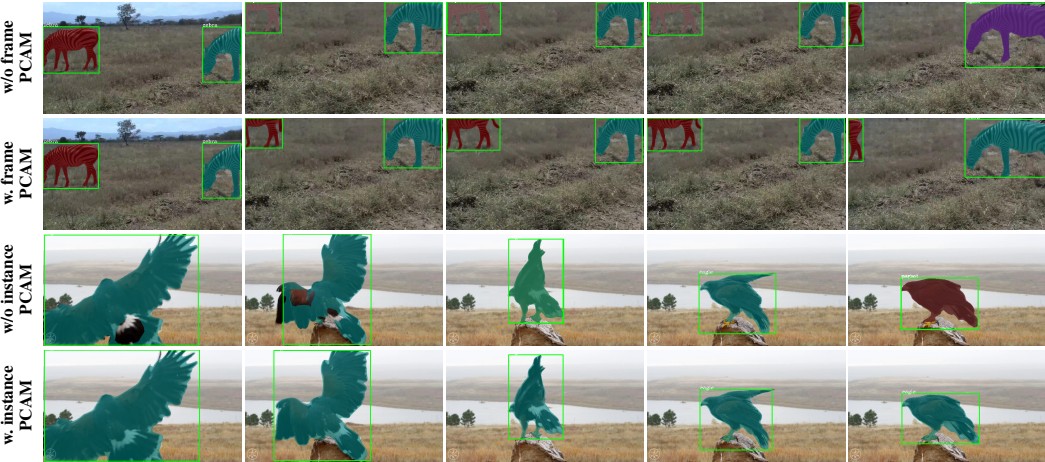

Figure 4: Qualitative impact of our PCAM on YouTube-VIS. Mask colors encode object identity. Our frame-level PCAM (second row) helps provide consistent detections and preserve identities compared to the baseline (first row). The instance-level PCAM (fourth row) provides more accurate masks, while further improving identity consistency compared to not employing our module (third row).

lengths in Table 3. A temporal length of 1 thus means that no prior temporal information guidance is used during video instance segmentation. By varying the frame length from 1 to 32, the mask AP increases from 32.5% to 35.4%, which reveals that richer temporal information with multiple views of a segmented object indeed brings more gain to model performance. For the number of frame-level prototypes, we used 64 during training and testing. The results on YouTube-VIS in Table 8 show that the precision saturates for larger numbers of prototypes.

**Effect of multi-layer temporal aggregation**  Since we perform temporal feature aggregation on the extracted FPN features, to help deal with objects with partial occlusion and large-scale variation, we also study the effect of using different levels of the extracted FPN features. In Table 4, we select the FPN feature map from P3-P5 layers for (excluding P6 and P7 due to impractical computation cost), and perform prototypical temporal aggregation on each FPN layer. We find that multi-layer information is also important to final model performance.

**Computation and memory efficiency**  In Table 5 we analyze different attention mechanisms. Compared to standard space-time memory reading using non-local attention [39, 28] or recent popular transformer [41, 6] with multi-head self-attention layer, the prototypical cross-attention with condensed prototypes not only enjoys high accuracy advantage, but also largely reduces the memory consumption and computation amount. For input tube length 8, the prototypical memory consumption is less than 10% of the transformer with negligible FLOPs computation due to the small number of representative prototypes in (5).

**Effect of instance-level prototypical appearance module**  We analyze the instance-level prototypical cross-attention module, which represents each video tracklet using the contrastive prototypes. In

Table 6: Ablation study on number of instance-level prototypes on YouTube-VIS.

| Pos. Proto. Number | Neg. Proto. Number | AP | $AP_{50}$ |
|---|---|---|---|
| 0 | 0 | 32.5 | 53.0 |
| 1 | 0 | 32.4 | 52.3 |
| 0 | 1 | 32.1 | 52.4 |
| 1 | 1 | 32.7 | 52.8 |
| 5 | 5 | 33.1 | 53.6 |
| 30 | 30 | **33.9** | **54.1** |
| 50 | 50 | 33.6 | 53.8 |

Table 7: Ablation on instance-level EM feature propagation and updating on YouTube-VIS.

| version | AP | $AP_{50}$ |
|---|---|---|
| No instance prototype propagation | 33.5 | 53.2 |
| Using initial instance prototype | 33.0 | 52.8 |
| Update momentum = 0.2 | **34.3** | **53.8** |
| Update momentum = 0.5 | 34.0 | 53.6 |

Table 8: Ablation on number of frame-level prototypes on YouTube-VIS.

| Proto. Number | AP | $AP_{50}$ |
|---|---|---|
| 8 | 32.6 | 52.8 |
| 16 | 33.1 | 53.3 |
| 32 | 33.9 | 53.5 |
| 64 | **34.2** | 53.7 |
| 128 | 34.1 | **53.8** |

Table 9: Results of varying EM iterations for our PCAN on YouTube-VIS.

| Iteration number | AP | $AP_{50}$ | $AP_{75}$ | $AR_1$ | $AR_{10}$ |
|---|---|---|---|---|---|
| 1 | 33.3 | 53.4 | 35.8 | 33.2 | 38.8 |
| 2 | 33.7 | 53.9 | 36.4 | 33.6 | 39.3 |
| 4 | 33.7 | **54.1** | 36.5 | 33.9 | 39.5 |
| 6 | **33.9** | 54.0 | **36.8** | **34.1** | **40.0** |
| 8 | 33.6 | 53.6 | 36.1 | 33.7 | 39.3 |

Table 6, we study the influence of instance prototype number and the effect of foreground-background contrasting. Using both positive and negative prototypes improves AP from 32.5% to 33.9%. Compared to the single prototype representation, the GMM demonstrate a stronger appearance modeling ability. We further find that the performance saturates when the number is larger than 60. In the Figure 6 and supplementary file, we provide additional instance cross-attention maps visualization to highlight the various attended regions.

In Table 7, we investigate the effectiveness of instance prototype (including the both positive and negative ones) propagation in an online manner, and compared it with using the instance prototype in the initial frame or current frame. We find that updating object prototypes recurrently with a momentum of 0.2 improves video segmentation AP of 1.3%.

**Influence of EM iteration number** We study the influence of EM iteration number $T$ during condensing prototypes and the results are shown in Table 9. Using temporal memory length 4, we find that the accuracy gains of PCAN increase with more iterations from 1 to 6, and the improvement starts to saturate when $T \geqslant 6$. We use the same iteration number during training and test.

**Ablation study on KITTI-MOTS** We also train PCAN on the KITTI-MOTS [37] training set and conduct ablations on the instance and frame PCAMs. In Table 10, PCAN with window size 8 on val set also shows significant improvements compared to the TrackR-CNN [37] (a two-stage tracker based on Mask R-CNN) on the benchmark. Note that many published methods on KITTI-MOTS, such as Vip-DeepLab [31], EagerMOT [17] and MOTSFusion [24], use 3D bounding boxes, LIDAR point clouds, or optical flow (PointTrack [44]). In contrast, our method only relies on RGB images.

**Qualitative analysis** In Figure 4, we showcase qualitative ablation results of PCAN on Youtube-VIS. Compared to the baseline, we see that our model results in more consistent segmentation and better tracking using prototypical cross-attention module. We also provide visual results on BDD100K in Figure 5, where PCAN produces robust tracking and segmentation results even under large object appearance change (first row) or low illumination (second row). In the 3rd row, PCAN has limitations in handling missing detections (the person in the first frame) with limited appearance information under extreme lighting, and produce tracking errors in the second frame when visible parts of the same car is totally different across frame and with low appearance similarity.

**Cross-Attention Visualization** In Figure 6, we visualize instance-level prototypical cross-attention of the interested car for both the corresponding foreground and background regions on three continuous frames on BDD100K, where the attended region of each object prototype reveals the implicit unsupervised temporal consistency. More visualization cases on instance and frame cross-attention maps and relevant analysis are in the supplementary file.

**Societal impact** PCAN has high potential impact in important applications, such as transportation, sports analysis, and self-driving vehicles. However, this powerful technology can be deployed in human monitoring and surveillance as well which raise ethical and privacy issues. Potential negative impact can be avoided by enforcing a strict and secure data privacy regulation such as the GDPR,

Table 10: Ablation study of PCAN on KITTI-MOTS [37] validation set.

| Method | Car-MOTSA | Ped-MOTSA | Car-MOTSP | Ped-MOTSP |
|---|---|---|---|---|
| TrackR-CNN [37] | 87.8 | 65.1 | 87.2 | 75.7 |
| PCAN w/o frame PCAM | 87.3 | 65.3 | 86.9 | 75.0 |
| PCAN w/o instance PCAM | 87.8 | 65.8 | 87.1 | 75.5 |
| PCAN (Ours) | **89.6** | **66.4** | **88.3** | **76.1** |

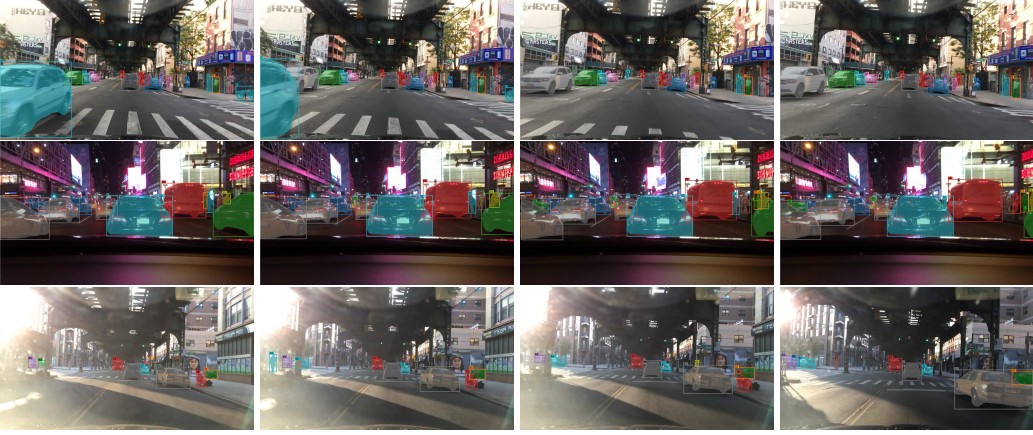

Figure 5: Qualitative results of our method on BDD100K. PCAN produces robust tracking and segmentation results under large motion and appearance changes (1st row) and heavy traffic in low-light conditions (2nd row). In the 3rd row, PCAN misses a detection (the person to the left in 1st frame), and produces tracking errors (2nd frame) when it covers totally different regions of the car with low appearance similarity. Zoom for better view. Video results are in the suppl. file.

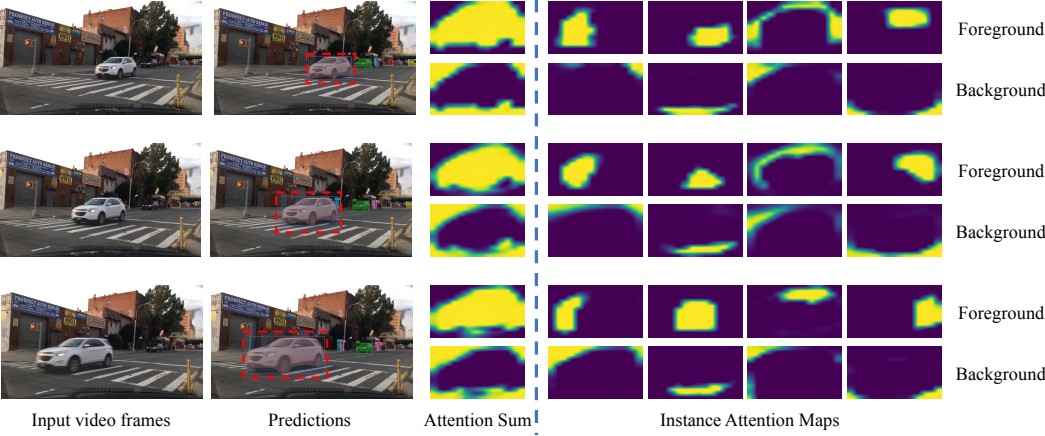

Input video frames    Predictions    Attention Sum    Instance Attention Maps

Figure 6: Instance cross-attention maps visualization for the car specified by the red dotted bounding box on BDD100K. We select the first four foreground/background prototypes as example, where each one focuses on specific car sub-regions with implicit unsupervised temporal consistency over time.

proper technology management education, and having an open dialogue among various stakeholders on how such technology should be deployed and regulated.

## 5   Conclusion

We present PCAN, a new online method for MOTS. PCAN first distills the space-time memory into a set of frame-level and instance-level prototypes, followed by cross-attention to retrieve rich information from the past frames. In contrast to most previous MOTS methods with limited temporal consideration, PCAN efficiently performs long-term temporal propagation and aggregation, and achieves large performance gain on the two largest MOTS benchmarks with low computation and memory cost. We validate the efficacy of PCAN on both the existing one-stage and two-stage trackers. We believe PCAN will significantly benefit more video understanding tasks in the future.

## Acknowledgments and Disclosure of Funding

This research is supported in part by the Research Grant Council of the Hong Kong SAR under grant no. 16201818 and Kuaishou Technology.

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
