# Supplementary Material:
# Prototypical Cross-Attention Networks for Multiple Object Tracking and Segmentation

**Lei Ke**[1,2]  **Xia Li**[1]  **Martin Danelljan**[1]  **Yu-Wing Tai**[3]  **Chi-Keung Tang**[2]  **Fisher Yu**[1]

[1]ETH Zürich          [2]HKUST          [3]Kuaishou Technology

{lkeab,cktang}@cse.ust.hk, {xia.li,martin.danelljan}@vision.ee.ethz.ch

yuwing@gmail.com, i@yf.io

In this appendix, we present additional experiment analysis, implementation details, prototypical cross-attention weights visualization (including instance-level and frame-level) and dataset license/safety. Please refer to the video on our project website (http://vis.xyz/pub/pcan) for the PCAN method introduction and video results on BDD100K [5] and Youtube-VIS [4].

## 1  Supplementary experiments

**Detailed results comparison on BDD100K**    Table 1 provides a more detailed class-wise comparison with competitive baseline method QDTrack-mots [3] on the BDD100K MOTS validation set. The temporal segmentation of PCAN improves MOTSA over the QDTrack-mots across all categories, especially for the more difficult classes such as motorcycle and bicycle due to the unbalanced class distribution of the training set. Also, PCAN effectively decrease the IDSw number of the car and pedestrian class for over 10%, where these two classes occupy the major part of the dataset.

Table 1: Class-wise performance comparison with strong baseline method QDTrack-mots [3] on BDD100K, which includes major MOTS metrics MOTSA, MOTSP and IDSw. - denotes NaN returned by the online evaluation server.

| Model | Metric | Average | Human | | | Vehicle | | | | | Bike | | |
|---|---|---|---|---|---|---|---|---|---|---|---|---|---|
| | | | Overall | Pedestrain | Rider | Overall | Car | Truck | Bus | Train | Overall | Motorcycle | Bicycle |
| QDTrack-mots | MOTSA↑ | 23.5 | 40.9 | 41.5 | 18.6 | 58.0 | 60.5 | 36.7 | 29.2 | 0.0 | 4.4 | -5.1 | 6.8 |
| **PCAN (Ours)** | MOTSA↑ | **27.4** | **43.0** | **43.6** | **19.7** | **59.9** | **62.5** | **36.9** | **33.9** | 0.0 | **14.5** | **5.9** | **16.7** |
| QDTrack-mots | MOTSP↑ | 66.3 | 75.2 | 75.3 | 66.8 | 84.3 | 84.2 | 85.3 | 86.2 | - | 69.5 | 62.6 | 70.1 |
| **PCAN (Ours)** | MOTSP↑ | **66.7** | **75.5** | **75.6** | **67.7** | **84.6** | **84.5** | **85.9** | **86.6** | - | **70.0** | **63.0** | **70.7** |
| QDTrack-mots | IDSw↓ | 973 | 387 | 385 | 2 | 585 | 556 | 28 | 1 | 0 | 1 | 0 | 1 |
| **PCAN (Ours)** | IDSw↓ | **876** | **345** | **343** | **2** | **531** | **501** | 29 | 1 | **0** | **0** | **0** | **0** |

## 2  More implementation details

We provide more implementation and training details of our Prototypical Cross-attention Network (PCAN) based on existing MOTS approaches for Youtube-VIS and BDD100K in this section.

**PCAN implementation and training on Youtube-VIS**    On Youtube-VIS, we build our segmentation tracker on the one-stage segmentation model [1]. For ablation study, our method is trained on four GPUs using ResNet-50, where we use SGD for optimization and set initial learning rate to 0.0025 with total batch size 8. We train PCAN for 12 epochs (taking about 8 hours with NVIDIA RTX 2080 Ti), and decrease the learning rate by 0.1 after 8 and 11 epochs. For leaderboard submission with ResNet-101, we double both the training batch size and learning rate using 8 GPUs and train the model for 16 epochs. The PyTorch data and model parallel scheme is used to speed up the training process of our model.

35th Conference on Neural Information Processing Systems (NeurIPS 2021).

Compared to previous real-time MOTS methods [1, 2] with limited temporal modeling (running at 20-30 FPS), although PCAN has additional space-time memory reading operation, it still runs at about 15 fps on ResNet-50 thanks to the the condensed clustering and efficient EM operation. Note that the EM algorithm has no trainable parameters, and we use 30 positive prototypes and 30 negative prototypes to represent each instance. During testing, the space-time memory is updated online by adding the latest frame embedding and mask predictions. During training, to guarantee the reference frame has a stronger spatial correspondence with the query frame, the sampling is done locally within $L_r$ frames, where $L_r$ denotes the temporal memory length.

**PCAN implementation and training on BDD100K**   On BDD100K, we build PCAN by extending the two-stage MOT method [3] with our prototypical temporal segmentation modules and use ResNet-50 as backbone. We train our models with a total batch size of 8 and an initial learning rate of 0.01 for 12 epochs on 4 gpus. We adopt the model pre-trained on BDD object detection set following the training strategy of [3] and use single-scale image size $1296 \times 720$ for both training and test. When conducting online joint object tracking and segmentation, we initialize a new track if its detection confidence is higher than 0.8. Note we employ prototypical instance attention on the RoI feature instead of the whole frame feature map as on one-stage tracker.

**Multi-level temporal feature aggregation**   We design multi-level temporal feature aggregation for frame-level prototypical cross-attention based on the FPN feature map from P3-P5 layers as shown in Figure 1. The temporal aggregation module takes all reconstructed frame feature from memory and current frame feature at corresponding FPN level and computes the linear fusion weights by the feature similarity to current frame prediction, where we show its effect in Table 4 of the paper.

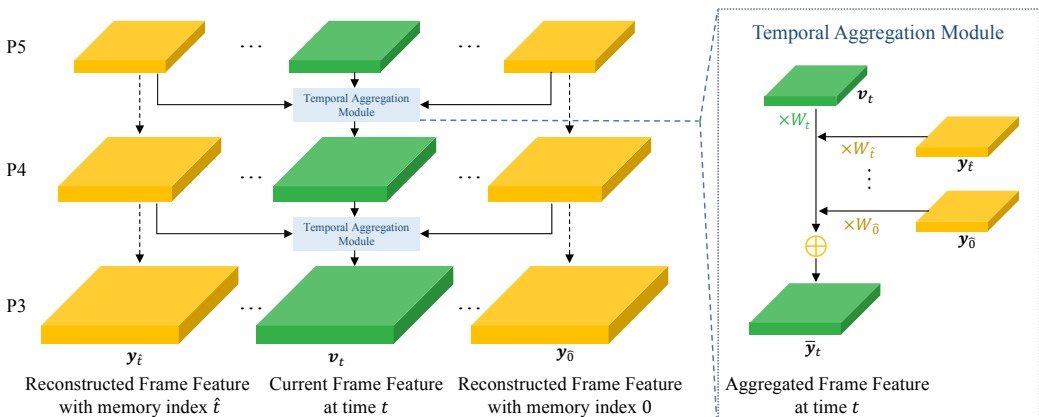

Figure 1: Multi-level temporal prototypical feature aggregation framework.

## 3   Prototypical Cross-attention weights visualization

To get a deeper understanding of our proposed prototypical cross-attention module, we visualize the cross-attention weights both in the instance-level and frame-level during the mask predictions.

**Instance cross-attention**   In Figure 2, we visualize instance-level prototypical cross-attention of the interested pedestrian for both the corresponding foreground and background regions on three continuous frames on BDD100K, where the attended region of each object prototype reveals the implicit unsupervised temporal consistency. In Figure 3, 4 and 5, we show more instance weight visualization on the Youtube-VIS with more semantically diverse objects.

**Frame cross-attention**   We show frame-level prototypical cross-attention maps in Figure 6, where we randomly select eight frame prototypes and show their attention distribution on the image. Obviously, each condensed prototype learn to correspond to some semantic concepts of the image covering both the foreground and background areas, such as human, skateboard, umbrella and tennis racket etc.

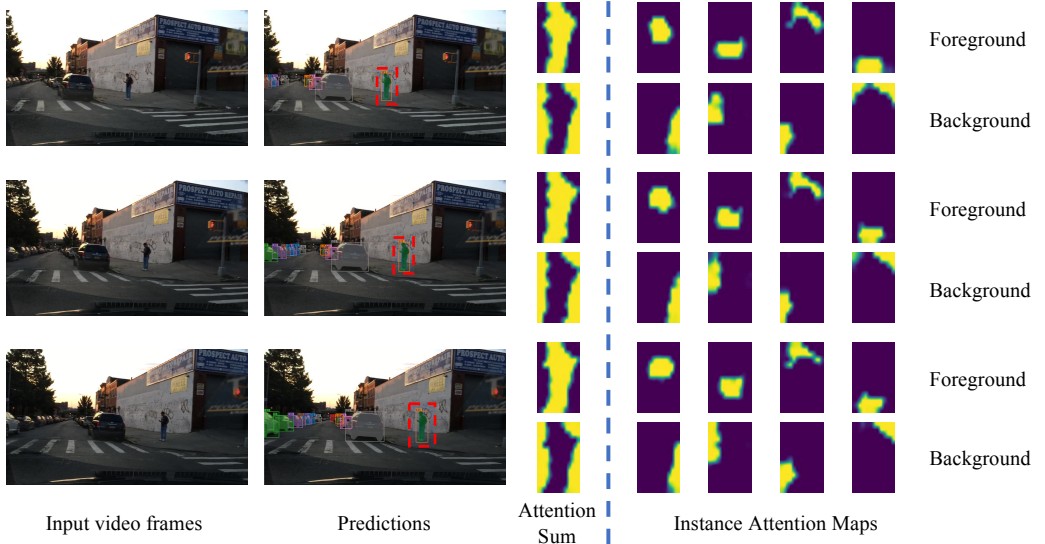

Input video frames | Predictions | Attention Sum | Instance Attention Maps

Figure 2: Prototypical instance cross-attention maps visualization for the pedestrian specified by the red dotted bounding box on BDD100K dataset, where the third foreground prototype consistently look at the head part over time while the fourth one attends to the feet region.

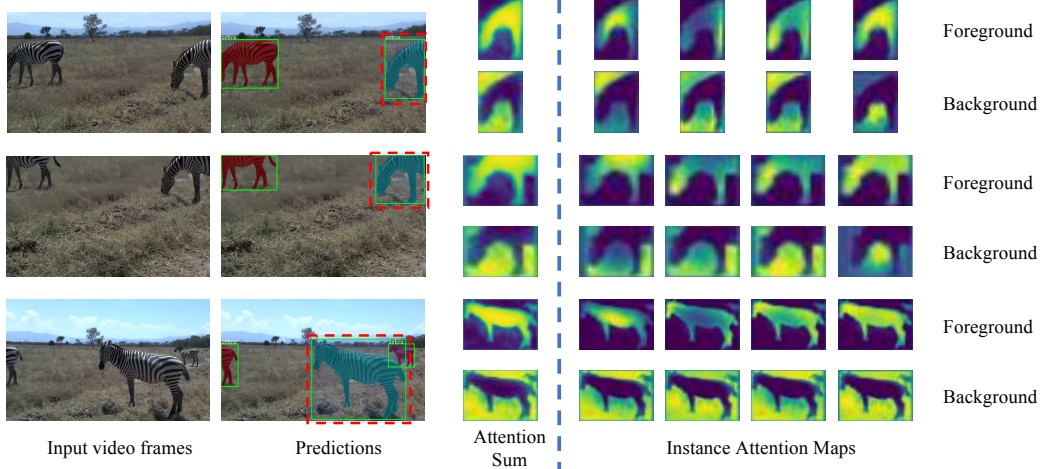

Input video frames | Predictions | Attention Sum | Instance Attention Maps

Figure 3: Prototypical instance cross-attention maps visualization for the zebra specified by the red dotted bounding box on Youtube-VIS dataset.

## 4    Dataset license and details

**Youtube-VIS.**    Youtube-VIS [4] is a public video instance segmentation dataset, consisting of semantically diverse images collected from video object segmentation dataset YouTubeVOS, including common objects such as person, animals and vehicles. We adopt the official dataset split for training and evaluation. Youtube-VIS is released for non-commercial research purpose only and ilcensed under a Creative Commons Attribution 4.0 License.

**BDD100K.**    BDD100K [5] is one of the largest public autonomous driving datasets, which contains diverse scene types such as city streets, residential areas, and highways. We adopt its official segmentation tracking set for model training and evaluation. BDD100K data is released under BDD License [1], which permits free academic usage.

---

[1]https://doc.bdd100k.com/license.html

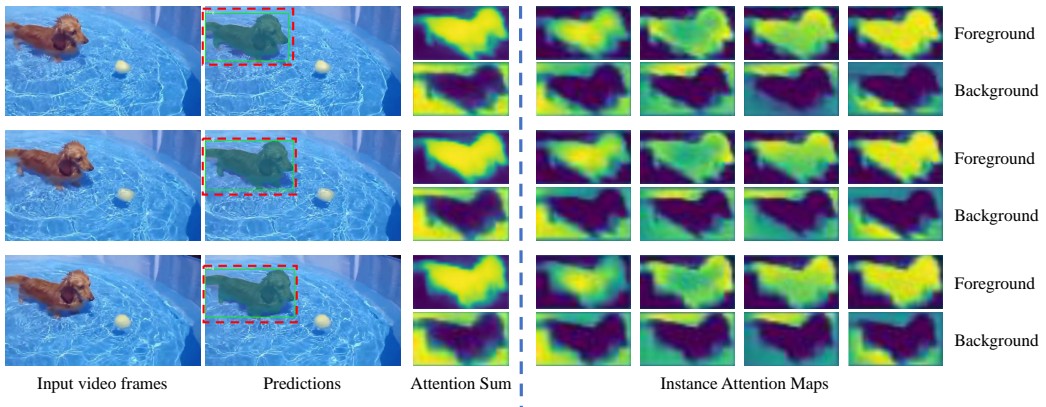

Figure 4: Prototypical instance cross-attention maps visualization for the swimming dog specified by the red dotted bounding box on Youtube-VIS dataset.

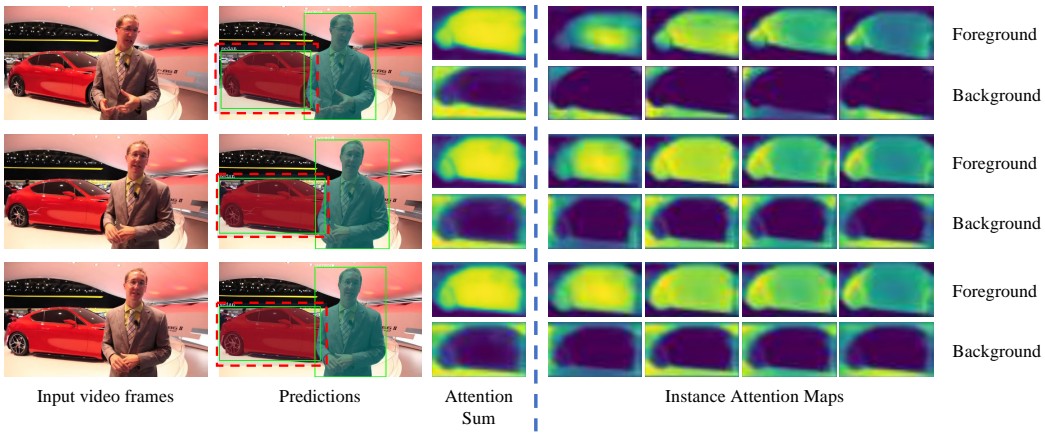

Figure 5: Protypical instance cross-attention maps visualization for the indoor car specified by the red dotted bounding box on Youtube-VIS dataset.

**Whether the datasets cover personally identifiable information or offensive content?** The Youtube-VIS and BDD100K are public video datasets. To best of our knowledge, no personally identifiable information or offensive content can be found. BDD100K is a real street scene video dataset which is for non-commercial use only. Even though BDD100K and Youtube-VIS cover the "person" class as one of the semantic annotation classes, no personally identifiable information or offensive content is found. Besides, the dataset creators had made a pledge that such content if ever exists and is reported will be immediately removed from their servers.

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

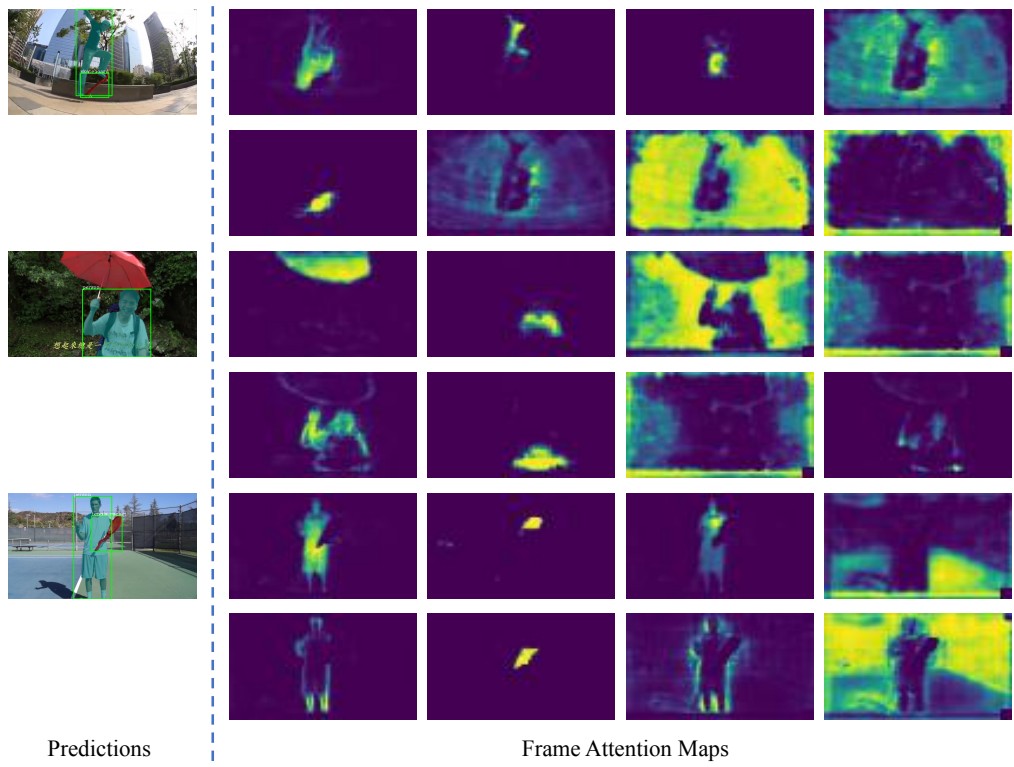

Predictions          Frame Attention Maps

Figure 6: Frame-level prototypical cross-attention maps on Youtube-VIS dataset, where we randomly select eight frame prototypes on each image for visualization.