# OpenReview forum: "Prototypical Cross-Attention Networks for Multiple Object Tracking and Segmentation"
_NeurIPS.cc/2021/Conference — NeurIPS 2021 Spotlight_

### Official Review · Reviewer_x3ez · 2021-07-15

**Rating:** 5
**Confidence:** 5

**Summary:**

This paper considers the problem of multiple object tracking and segmentation.  The overall framework is a memory-based cross-attention architecture, using temporally previous reference features to correlate and then enhance the current predictions. The main contribution is two-fold. First, they propose to use GMM to compute several clustering centers for the memory features as the "prototype memory", so space and time cost of cross-attention is largely reduced, and using longer temporal reference scope is then possible. Second, they propose an instance-level memory to contrast foreground and background and also allow moving average updating of the template.  Experimental results show promising tracking accuracy.

**Limitations And Societal Impact:**

I do not see any negative societal impact of this work.

**Main Review:**

Appreciate the authors for their effort! Overall this is a technically solid paper, with a clear motivation (save the space and time cost of memory & attention), presents a complete solution, and conducts extensive experiments to validate the effectiveness of the proposed method.  The idea makes sense and the results seem good. However I have the following concerns, if they are properly addressed I would consider raising my rating.

1. Considering using attention-based models and space-temporal memory has already been a most popular framework for MOTS/VIS,  I believe the biggest contribution of this work is that the authors propose the concept of "memory prototype" to reduce the size of memory and thus enable longer temporal reference scope. However it is not shown at what cost the improvement of accuracy is achieved. An elephant in the room is that almost all implementation details of the GMM clustering are missing, e.g. how often to perform the clustering,  and how long one clustering step takes, how to choose the GMM component number $N$, and how the performance will vary with different $N$.

2. Regarding temporal feature aggregation. I notice that in the proposed method each reference frame is first aligned to spatially match the target frame, and then a weighted sum is used to aggregate reference features along the temporal dimension in each spatial location.  This makes sense, but the necessary should be clarified because there exists a simpler alternative: concatenating all the reference features (from multiple reference frames) into a *single* reference feature set and apply cross attention between this single reference feature set and the target features. In fact, several existing methods[a,b] adopt this strategy, and the benefit is that no explicit temporal aggregation is needed.

3. I am a bit confused that why both frame-level and instance-level memory are needed. It looks like they have the exact same formulation, and the only difference is that the instance-level one supports moving average updating for the foreground template. Is it possible to use instance-level memory only?

4. Regarding tracklet feature propagation and updating, Eq (8), how do you ensure the temporal consistency of the GMM components? For instance, it is possible that the component $j$ in timestep $t-1$ does not correspond to the component $j$ in timestep $t$. Do you do association here?

5. Again, the comparison in Table 5 is not fair. It makes less sense to compare FLOPS, could you compare the runtime directly, between the proposed method (including the runtime of GMM) and Non-local attention?

6. In the supplementary video, I see several visualizations but found them usually too short (2-5 sec.). If it is possible, could the authors provide more results of longer video sequences?  It is important to see how a tracker performs in longer videos.

7. Minor things:

    - L#48 "a MOTS approach" -> "an MOTS approach"

    - L#146, should $k_{\hat{t}i}^{\mu}$ be $k_{\hat{t}j}^{\mu}$?

    - L#171, "GMM (2)": does it mean GMM (Eq. 2)?

a. MAST: A Memory-Augmented Self-Supervised Tracker. https://arxiv.org/pdf/2002.07793.pdf

b. Space-Time Correspondence as a Contrastive Random Walk. https://arxiv.org/pdf/2006.14613.pdf

**Time Spent Reviewing:**

2

---

> ### Author Response · Authors · 2021-08-10
> **Response to Reviewer x3ez**
>
> Thank you for acknowledging our work being technically solid with clear motivation and validated in its effectiveness. We address your points below.
>
> **Q1. The details of the GMM clustering.**
>
> We adopt the Expectation-maximization (EM) algorithm for clustering the memory features (L119). We use 6 EM iterations (L13-L15 of the Supp.) for GMMs. It takes about 3 ms per frame, corresponding to 4.5% of the total computation. This is a small cost compared to the substantial performance improvements brought by our approach.
> We analyze the number of GMM components for the instance-level PCN in Table 6. We use 30 positive and 30 negative prototypes (L30 of supp.). For the number of frame-level prototypes, we used 64 during training and testing. To analyze the impact of this parameter on YouTube-VIS, using a temporal memory length of 8. The results below shown consistent performance increase with gradual saturation for larger values.
>
> | Num. Frame Prototypes | 8 | 16 | 32 | 64 | 128 |
> | --- | --- | --- | --- | --- | --- |
> | AP |  32.6 |  33.1  | 33.9 | 34.2 | 34.1 |
> | AP$_{50}$ |  52.8 | 53.3 | 53.5 | 53.7| 53.8 |
>
>
> **Q2. Temporal feature aggregation strategy.**
>
> We design the temporal feature aggregation strategy to further reduce the computational and memory impact of our PCN. While all reference features could be concatenated, it would require much memory and significantly larger computational cost. The memory needs to be re-clustered after each update, which is costly when storing many frames. In contrast, our solution only needs to store the past condensed memory prototypes, such that we can use longer temporal information. It allows the computed prototypes of the past frame to be efficiently reused.
>
> **Q3. The use of both frame-level and instance-level prototypes.**
>
> We summarize the impact of the frame-level and instance-level prototypical attention on YouTube-VIS in a single table below (results are found in Table 1, 3, and 6 in the paper). Both modules bring significant gains in final performance. The general nature of our PCN module allows us to apply it on both levels. The frame-level PCN can improve detection, classification, and association by aggregating temporal information in an earlier stage. The instance-level PCN then focuses on fine-grained temporal information for each instance in order to refine the final segmentation mask.
>
> | Method | AP | AP$_{50}$ |
> | --- | --- | --- |
> | Baseline |  32.5 |  53.0  |
> | Baseline + Frame-level PCN |  35.4 | 53.8  |
> | Baseline + Instance-level PCN |  33.9 | 54.1  |
> | Baseline + Frame & Instance-level PCN |  36.1 | 54.9  |
>
> **Q4. Details on the tracklet feature propagation/updating.**
>
> To achieve temporal consistency of GMM components, we use the stored instance-level prototypes from the previous frame as the initialization for the EM algorithm for the current frame. The resulting temporal consistency in the corresponding instance attention weights produced by these prototypes is seen in Fig. 3 of the paper, and Figs. 2-6 of the Supp. file.
>
> **Q5. The running time comparison between Prototypical cross-attention and non-local attention.**
>
> We compare the average running time per frame, and the performance of our method by using prototypical cross-attention and standard non-local attention for window length 8 of Youtube-VIS below. It shows replacing non-local attention with prototypical cross-attention has a minor influence on the inference speed while substantially reducing the memory consumption (see Table 5 of the paper).
>
> |    | Prototypical Cross-attention-based | Non-local Attention-based  |
> | --- | --- | --- |
> | Time (ms) per frame | 63.2 | 61.8 |
> | AP | 34.2 |  33.6 |
>
> **Q6. Longer video results in the supplemental file.**
>
> While our supplemental videos contain diverse driving scenarios, our video submission was restricted by the video size limitations of NeurIPS OpenReview. We will compress the video, and update it with longer sequence results.
>
> **Q7. L#48, L#146 and L#171.**
>
> Thank you and we will correct the typo mistakes. For L#171, it’s the same notation.
>
> **Q8. Discussion on the mentioned works [a] and [b].**
>
> We will add discussion on space-time memory usage of [a] and [b] in our related work section. However, please note that [a, b] are contributions to semi-supervised video object segmentation in a self-supervised manner, while PCN focuses on MOTS/VIS with our core contribution centered around the design of an effective prototypical attention module based on GMMs for efficiently utilizing long-term spatio-temporal video information.

---

> > ### Comment · Reviewer_x3ez · 2021-08-29
> > **Thanks for the response**
> >
> > Thanks! Most of my concerns are addressed in the response.
> >
> > Below I summarize my understanding: From Table 5 in the main text and Q5 in the response, I find that the advantages of the proposed method over the vanilla non-local attention are that 1) it saves a large amount of memory (9631-> 1419 MB when tube length=8), and 2) the performance is better (33.6->34.2 AP). But the running speed is not faster, and actually a bit slower due to the calculation of EM (If I am wrong please kindly point out).
> >
> > Here are my concerns regarding the above point,
> >
> > 1. Since the proposed method significantly reduced the memory demand, given a fixed memory (say 12 or 24 GB), what is the best performance that can be achieved by PCN and the vanilla non-local attention by enlarging the tube length? Maybe that will be a piece of good evidence to show PCN's benefit.
> >
> > 2. To me the improvements upon vanilla non-local attention is trivial somehow (0.5-0.6 AP).  However, this would not be important if the authors could make the first point I mention above.

---

> > > ### Author Response · Authors · 2021-08-29
> > > **Thanks for the suggestion on performance comparison given limited memory.**
> > >
> > > Thank you for the summarization and suggestion on performance comparison given limited memory.
> > >
> > > We reported the results of PCN using longer tube length 16 and 32 in Table 3 of the paper, where PCN further achieved 34.6 AP and 35.4 AP. Note that the largest video length of annotated frames in Youtube-VIS dataset is 36, and we found that bigger window size 36 doesn’t provide additional benefits compared to 32.
> > >
> > > For prototypical cross-attention of tube length 32 (memory consumption 5813M with 35.4 AP), it is still within the fixed memory 12GB but PCN has an obvious advantage of 1.8 AP compared to the vanilla non-local attention with tube length 8 of 33.6 AP (memory consumption 9631M in Table 5 of the paper).
> > >
> > > We will clarify this advantage of PCN in the revised version.

---

### Official Review · Reviewer_nMFp · 2021-07-16

**Rating:** 6
**Confidence:** 4

**Summary:**

This paper proposes a new prototypical cross-attention network for online multi-object tracking and segmentation. Specifically, the proposed method learns a set of foreground and background prototypes from the past frames. The learned prototypes are merged with the current frame embedding via cross-attention. Several experiments conducted on Youtube-VIS and BDD100K datasets show that the proposed method performs favorably against the existing methods.

**Limitations And Societal Impact:**

1. In Table 2, the method is evaluated on the BDD100K val set. What about the performance on the test set? Besides, how did you obtain the results of the compared methods? The results are different from the reported scores in Table 2 of the work [24].
2. The running speeds of the compared methods are not reported in Table 1.
3. The authors should summarize how to select some important parameters in the network in Section 4.1, e.g., the number of prototypes, and the update rate for prototypes.


**Main Review:**

1. The idea of the proposed method is interesting.
2. Extracted from the past frames, the prototypes in the frame-level and instance-level are effective to enhance the feature embeddings efficiently.
3. The ablation studies are conducted to demonstrate the effectiveness of the frame and instance prototypes.
4. The prototypical cross-attention module uses much less memory than the standard transformers.


**Time Spent Reviewing:**

8

---

> ### Author Response · Authors · 2021-08-10
> **Response to Reviewer nMFp**
>
> Thank you for acknowledging that our method is interesting and validated in both effectiveness and efficiency. Here we respond to the insightful comments.
>
> **Q1a. The evaluation results on the BDD100K test set.**
>
> Results on the BDD100K test set are reported below. Our approach consistently outperforms QDTrack-mots by a large margin.
>
> | Method | mMOTSA |mMOTSP | mIDF | ID sw. |
> | --- | --- | --- | --- | --- |
> | QDTrack-mots |  26.0 | 65.4  | 45.1 | 1154 |
> | PCN | 31.8  | 66.1 | 49.9 | 817 |
>
> We will add the corresponding results and related discussion in our final version. We obtained the evaluation result of compared methods by running their official code on the validation set.
>
> **Q1b. Why is the result different compared to the reported scores in [24].**
>
> BDD100k officially extended the validation set size this year (from 10 validation video sequences to 32), which resulted in the different reported results compared to [24]. And the ground truth (GT) mask annotation has been redefined to replace polygons with more accurate bitmasks.
>
> **Q2. The running speed of the compared methods in Table 1.**
>
> We compare PCN with SG-Net on Youtube-VIS using ResNet50 as below. Note that STMask and SG-Net have no space-time memory reading operations, but instead focus on joint optimization of different task heads, reducing box proposal number and better mask prediction in sub-regions. These contributions are complementary to ours and could be integrated to further reduce the computational cost of our PCN.
>
> | Method | AP | FLOPs | FPS  |
> | ---- | ---- | ---- | ---- |
> | STMask (CVPR’21) | 33.5 |  58.9 G  |  28  |
> | SG-Net (CVPR’21) | 34.8 |  69.6 G  |  23  |
> | PCN (Ours) | 36.1 | 92.7 G | 15 |
>
> We would add more detailed comparisons on FPS and FLOPs in the revised version.
>
> **Q3. Important hyperparameters selection, such as number of prototypes and update rate.**
>
> In Table 6 of paper, we experiment the influence of instance prototypes number to final performance, where 30 positive and 30 negative prototypes are selected (specified in L30 of the supp. file) because the results saturate afterward. The experiment of update rate is in Table 7 of the paper, where we show the performance under update rate 0, 0.2, 0.5 and 1.0. The value 0.2 is selected according to the result.
>
> For the number of frame-level prototypes, we used 64 during training and testing. To analyze the impact of this parameter, we evaluate its impact on YouTube-VIS using a temporal memory length of 8. As shown below, the results on YouTube-VIS also show that the precision saturates for larger numbers of prototypes and is stable.
>
> | Num. Frame Prototypes | 8 | 16 | 32 | 64 | 128 |
> | --- | --- | --- | --- | --- | --- |
> | AP |  32.6 |  33.1  | 33.9 | 34.2 | 34.1 |
> | AP$_{50}$ |  52.8 | 53.3 | 53.5 | 53.7| 53.8 |

---

### Official Review · Reviewer_h7Rc · 2021-07-16

**Rating:** 5
**Confidence:** 3

**Summary:**

This paper concerns the problem of multi-object tracking and segmentation, using a prototype-based Cross-attention Network (PCN). The model first extracts temporal and instance-wise prototypes using a GMM clustering for all objects of interest from the past frames. Then, it uses those prototypes and their aggregation to detect, associate, classify and segment each object of interest in the current frame. The results have been evaluated on two datasets,   Youtube-VIS and BDD100k, and have been compared against tracking and/or video segmentation frameworks.

**Limitations And Societal Impact:**

Yes, the authors have well addressed the limitation and potential negative social impact of their work.

**Main Review:**

1- Clarity: The paper is well written, and the idea is clearly explained. The technical section can be followed easily. However, the application focus of the proposed technique is not very clear to me. The main claim in the paper is that it is a framework for multi-object tracking (MOT) and segmentation. Considering the dataset used for the evaluation, I am not convinced that the framework is suitable for the MOT/MOTS problem (See the extra notes at the end of my comments).

2- Contribution: the paper does not offer any new theoretical contribution. It is rather a pure application paper using built upon the existing and well-explored machine learning ideas (cross attentions, prototypical generations and contrastive learning) with some tweaks and minor extensions to address a challenging practical problem (i.e. Multi-object tracking and segmentation paper). While the application papers are also well acknowledged by the NeurIPS research community, but such papers require substantial experimental evaluation on the existing benchmark datasets to validate the efficacy of their proposed system. Without this, I would be hesitant to vote for the acceptance of such a paper in this venue. This would be my major concern about this paper elaborated Next.

3- Experiments and datasets:  The experiments are tested on two datasets one could be assumed suitable for MOTS problem (BDD100k), but the other one is mainly for video object segmentation (VOS) problems. Considering the main claim of the paper about addressing MOTS problem and taking this into account that this is an application paper, I am not sure why the superiority of this approach has not been tested on well-established MOTS benchmark such as MOTSChallenge and KITTI (MOTS).

4- Some missing details: I didn't find how the approach deals with track initiation and termination and occlusion handling in this framework.


Extra notes: The challenges in the MOT problem is very different from what is in visual object tracking (VOT) or video object segmentation (VOS).  In MOT, the goal is to track many similarly looking instances (often with the identical semantic class, e.g. pedestrians, Cars ), while in VOT or VOS, the objects of interest are often semantically well distinguishable. Otherwise, similar-category objects are well separated (no need to deal with challenges in MOT). The MOT methods need to be robust in fine-grain visual similarities and they require reliable data association, track management techniques, while VOS and VOT require more general but robust visual representation.  There is neither experimental evidence nor theoretical insight this framework can be robust for all these problems



**Time Spent Reviewing:**

3

---

> ### Author Response · Authors · 2021-08-10
> **Response to Reviewer h7Rc**
>
> We thank the reviewer for acknowledging that our paper is well-written with a clear idea description. Here we respond to the insightful comments and questions.
>
> **Q1,Q3. MOTS/VIS vs VOS and corresponding evaluation datasets.**
>
> VOS and MOTS/VIS are inherently different problems. We address the MOTS/VIS task, which requires joint detection, classification and tracking of objects. In contrast, VOS aims to propagate the mask, given in the first frame, of an object of unknown category. Youtube-VIS is different from the Youtube-VOS dataset. Specifically, Youtube-VIS requires detecting, classifying, segmenting, and tracking objects from 40 different categories, without any given initial mask.
>
> We have evaluated PCN on two large-scale MOTS/VIS benchmarks, where BDD100k covers the self-driving scenario and Youtube-ViS has more object categories. BDD100K Segmentation Tracking and KITTI-MOTS are both self-driving benchmarks, but BDD100K is 6 times larger (3,0817 vs 5,027 training images, 480K vs 26K instance masks) with richer and more diverse driving scenarios (shown in the supp. video). Note that many published methods on KITTI-MOTS, such as Vip-DeepLab, EagerMOT and MOTSFusion, use 3D bounding boxes, point clouds, or optical flow (PointTrack), whereas our method only relies on RGB images.
>
> For comparison, we train PCN on the KITTI-MOTS training set and evaluate it on the val set. Our initial results, with no hyperparameter tuning and window size 8, also show significant improvements compared to TrackR-CNN (a two-stage tracker based on Mask RCNN).
>
> | Method | Car-MOTSA | Ped-MOTSA | Car-MOTSP | Ped-MOTSP |
> | --- | --- | --- | --- | --- |
> | PCN | 89.6 |  66.4 | 88.3 | 76.1 |
> | PCN w/o frame prototypes |  87.3 | 65.3 | 86.9 | 75.0 |
> | PCN w/o instance prototypes |  87.8 | 65.8 | 87.1 | 75.5 |
> | TrackR-CNN | 87.8 | 65.1 | 87.2 | 75.7 |
>
> We will include these analyses in our next revision.
>
> **Q2. Contributions.**
>
> To the best of our knowledge, we are the first to propose a prototypical cross-attention mechanism, by performing GMM-based clustering of the space-time memory. Our approach addresses the well-known computational and memory drawbacks of the standard attention operation, allowing the efficient use of more long-term temporal information. While our work focuses on MOTS/VIS, the proposed module is generic and can be applied to other video understanding tasks.
>
> **Q4. The details on track initiation and termination and occlusion handling.**
>
> We initialize a new track when its detection confidence score is higher than 0.8 (L41 of Supp.), and terminate the track when there are no matching objects in the next 30 frames. PCN implicitly handles occlusions by attending to older frames in the long-term spatio-temporal memory and instance feature propagation. Figure 4-5 and the supplementary video show our approach can provide more consistent detections and classifications, and preserve instance identities using long temporal information.

---

### Official Review · Reviewer_q7JA · 2021-07-16

**Rating:** 5
**Confidence:** 4

**Summary:**

This paper proposes to address the problem of multiple-object tracking and segmentation (MOTS) by creating a space-time memory to store useful information from previous frames that can be accessed with an attention module. To keep the computational complexity manageable, information is summarised in "prototypes" represented by a Gaussian Mixture Model.

Large-scale experiments are performed on the YouTube-VIS and BDD-100K benchmarks.

**Limitations And Societal Impact:**

The authors haven't addressed potential societal impacts, but I don't think this constitutes a problem for the method proposed. To respect the spirit of NeurIPS program chairs guidelines, a few lines could be dedicated to consider negative impacts such as empowering surveillance systems.

**Main Review:**

**Strengths**

I think that the main strength of the paper is to show the feasibility of applying a fairly low-cost attention module to the problem of MOTS. It's interesting to see that using prototypes in a space-time memory is a viable option for this type of problem.

**Weaknesses**

I list my main concerns below. For specific points, with references to line numbers, please see the next section.

* The paper does not reference relevant literature on a very similar problem: self-supervised learning from video (which is generally evaluated with single-object tracking and segmentation). In particular, methods like "MAST: A Memory-Augmented Self-supervised Tracker" also consider a memory and address similar issues to the ones discussed in this paper.
* I did not find any details on what is the loss used by the proposed method.
* Evaluation is done on benchmarks validation seta. This is fine, but hyperparameter choices should be detailed to show that are not chosen ad-hoc for the set used to report the results.
* Since one of the main strengths is the computational advantage with respect to other attention strategies, authors should also show how the computational complexity of the proposed methods fairs against non attention-based approaches (from Table 1 and 2)
* The presentation could be improved.

**Specific points**
* [L15-L16] MOTS and VIS have minor differences and respective benchmarks cover different scenarios. It would be good to detail these (perhaps in the experimental section). Also, in L67 the wording changes slightly and contradicts L15-16.
* [L23] Methods that take into account longer time spans do exist (MAST is the first that comes to mind, but I am sure there are  more).
* [L37] "Noise-reduced". Seems difficult to prove that this features are actually "noise-reduced"? What does it mean? With respect to what?
* [L37] The sentence seems disconnected from the sentence in the previous page.
* [L47] The list of contributions reads more like a breakdown of the paper structure rather than the actual contributions.
* [L57] "Regress". Predict?
* [L58] Not sure to understand "predicts a linear combination of mask bases".
* [L90] Is the proposed method also doing *detection*, or is it using existing detectors for that? It is not clear from this sentence.
* [L110] For the sake of clarity, it would be important at this point to better explain what can be an "item" in the memory and at what granularity (e.g. frame-level, instance-level, both, ...). As the text currently stands, it becomes clear only from L137.
* [L114] I would remove the use of qualifiers like "principled", "elegant", and so on in the text. It might well be true, but the text reads definitely better without them.
* [L124] What is the value of sigma? How did the authors choose it?
* [Fig3 caption] "Memorial" means "intended to commemorate someone or something". Not the appropriate usage here.
* [L148] Unclear what "parallel" means here. Do they use two different sets of weights?
* [L170] The fact that the masks are provided by a pre-trained detector could be made clearer earlier on.
* [Sec 3.3.1] Why rewriting the formulae of the previous page here? Could presentation be organised a bit differently to avoid that?
* [Sec 3.3.2] Many emphatic words that to me should be avoided: "compact and robust", "empower", "rich and dynamic", "naturally learn", "rich and valuable information". It's not super important, but I would stick to a more factual and dry writing style.
* [L185] What is the value of lambda, how did the authors chose it?
* [Sec 3] I could not find an explanation of how the loss is computed.
* [L212] Authors should explain what 1-stage and 2-stage frameworks are.
* [Table 3] This could be shown as a plot, and it would be good to go beyond 32 to show where things start to break.
* [Table 4] Why not reporting also P4-P5 and P3-P4-P5?

**Time Spent Reviewing:**

5

---

> ### Author Response · Authors · 2021-08-10
> **Response to Reviewer q7JA**
>
> Thank you for your review. Here we respond to the insightful comments.
>
> **Q1.  Missing related literature in self-supervised learning from videos, such as MAST.**
>
> We will add the discussion on space-time memory usage of MAST in our related work section. MAST is a semi-supervised video object segmentation (VOS) method, while PCN focuses on MOTS/VIS. Our core contribution is the prototypical attention module for efficiently utilizing long-term spatio-temporal video, while MAST focuses on self-supervised learning.
>
> **Q2. The loss computation and details.**
>
> We employ the standard segmentation, detection and classification losses in the Mask R-CNN (L20 of supp.) and SipMask (L36 of supp.) baselines on BDD100K and Youtube-VIS respectively. For object association, the cross entropy loss is used for the tracklet to be classified as one of the previously appeared instances, or as a new instance. We will add further details in the supplementary.
>
> **Q3. Hyperparameter choices.**
>
> We analyze the effect of hyper parameters in Table 3, 5, 6, and 7 of the main paper. We found the results to be stable wrt these hyperparameters, achieving consistent gain when increasing memory length (Table 3) or number of prototypes (Table 6).
>
> Since the Youtube-VIS test set is only available during the challenge phase, we evaluate on the validation set (also only accessible through an online server). To further validate our approach, we here provide results on the BDD100k test set (without altering any hyper parameters). Our approach consistently outperforms the QDTrack-mots method by a large margin. We will add this result in our final version.
>
> | Method | mMOTSA |mMOTSP | mIDF | ID sw. |
> | --- | --- | --- | --- | --- |
> | QDTrack-mots |  26.0 | 65.4  | 45.1 | 1154 |
> | PCN | 31.8  | 66.1 | 49.9 | 817 |
>
>
> **Q4. Computational complexity comparison with non-attention based methods.**
>
> We compare PCN with ST-Mask and SG-Net on Youtube-VIS using ResNet50 as below. These two methods focus on joint optimization of different task heads or reducing box proposal number to improve efficiency, where their contributions are complementary to ours and could be integrated to further reduce the computational cost of our PCN. We will add more detailed comparisons in the revised version.
>
> | Method | AP | FLOPs | FPS  |
> | ---- | ---- | ---- | ---- |
> | STMask (CVPR’21) | 33.5 |  58.9 G  |  28  |
> | SG-Net (CVPR’21) | 34.8 |  69.6 G  |  23  |
> | PCN (Ours) | 36.1 | 92.7 G | 15 |
>
> **Q5. Presentation.**
>
> We will revise the manuscript according to the reviewer’s detailed suggestions.
>
> Specific points:
>
>
> * [L37] “Noise-reduced”: We mean that noisy and non-robust information in the extracted image features are averaged out when condensed into prototypes. As shown in table 5, directly using the raw features with non-local attention leads to inferior results. This strongly indicates the added robustness provided by our prototypes.
>
> * [L58] “linear combination of masks” means the one-stage model predicting a set of coefficients to compute the weighted sum of the mask templates, which is discussed in detail in YoloAct [3].
>
> * [L90] Yes, our approach also performs detection by extending the baseline single-frame detectors SipMask and Mask R-CNN respectively. See L202-L203 in the paper and Section 2 of the Supp. file.
>
>
> * [L124] We simply set sigma to 1 and let the network learn the scale of the underlying embedding space used for clustering.
>
>
> * [L148] “Parallel”: In Fig. 2, the input frame features are fed into these two different conv layers in parallel to produce key and value embedding.
>
>
> * [L185] For lambda, we use 0.2 according to the experiment result in Table 7, where we compare under lambda values 0.0 (second row), 0.2, 0.5 and 1.0 (top row).
>
>
> * [L212] Two-stage denote detection paradigms following R-CNN, where a proposal generation step is followed by RoI pooling and prediction. A one-stage model is proposal-free.
>
> * [Table 3] The largest video length of annotated frames in Youtube-VIS dataset is 36. We found that bigger window size 36 doesn’t provide additional benefits. Also, we report results by setting window sizes in incremental steps of base 2 exponentiation.
>
>
> * [Table 4] P3-P5 in Table 4 denotes all levels from P3 to P5, including P4. As shown in the table, high-resolution features in P3 are important for accurate segmentation. We will nevertheless add P4-P5 in the revised version.
>
> **Q6. Negative social impact.**
>
> Please refer to L272-277 in the paper. As agreed by Reviewer h7Rc, “the limitation and potential negative social impact of the work has been well addressed”.

---

### Decision · Program_Chairs · 2021-09-28

**Decision:**

Accept (Spotlight)

**Comment:**

The paper received borderline negative reviews, three "below threshold" and one "above threshold". The main concerns were:

1. missing lit review on self-supervised methods.
2. choice of hyperparameters, since evaluation is only on the validation set.
3. missing evaluation on the BDD100k test set.
4. limited theoretical contribution. application paper, but not enough experiments.
5. missing details, clarity.

The authors provided a detailed response. The reviewers were only partially satisfied with the response. In particular, reviewers noted that: 1) low novelty and insight - the paper is combination of existing techniques, with no particular insight gained; 2) discussion of the relationship with self-supervised models was not sufficient in the paper or the response; 3) although the memory is saved, the computation cost is not fully reported or analyzed; 4) other missing details.  In the end, all reviewers were negative. The AC agrees and also recommends reject.

**Consistency Experiment:**

NeurIPS has a long history of experimentation. In 2014, NeurIPS ran an experiment in which 10% of submissions were reviewed by two independent committees to quantify the randomness in the review process. This year, we repeated a variant of this experiment to see how the quality of the review process has changed over time.  This paper was part of the experiment and was therefore assigned to two committees (consisting of reviewers, an Area Chair, and a Senior Area Chair) that reached independent decisions.  If both committees made the same recommendation, this recommendation was followed. If a single committee recommended acceptance, the paper was accepted (with the exception of a few cases in which the other committee identified what we considered a fatal flaw, e.g., an error in a key result).

This copy’s committee reached the following decision: **Reject**

The other committee assigned to the paper recommended **Accept (Spotlight)**.  You can find the other set of reviews, along with any follow up discussion with the authors here:
https://openreview.net/forum?id=OkFPq7ZtsQ